

# Is autumn the key for dengue epidemics in non endemic regions? The case of Argentina

Anibal E. Carbajo[1,2], Maria V. Cardo[1,2], Pilar C. Guimarey[3], Arturo A. Lizuain[3], Maria P. Buyayisqui[4], Teresa Varela[4], Maria E. Utgés[3], Carlos M. Giovacchini[4] and Maria S. Santini[3]

[1] Universidad Nacional de San Martin, Instituto de Investigacion e Ingenieria Ambiental, Laboratorio de Ecologia de Enfermedades Transmitidas por Vectores, General San Martin, Buenos Aires, Argentina
[2] Consejo Nacional de Investigaciones Cientificas y Tecnicas, Argentina
[3] Centro Nacional de Diagnóstico e Investigación en Endemo-epidemias CeNDIE - ANLIS, Ministerio de Salud de la Nacion, Buenos Aires, Argentina
[4] Área de Vigilancia de la Salud, Dirección de Epidemiología, Ministerio de Salud de la Nación, Buenos Aires, Argentina

Corresponding author
Maria V. Cardo,
mcardo@unsam.edu.ar

## ABSTRACT

**Background**. Dengue is a major and rapidly increasing public health problem. In Argentina, the southern extreme of its distribution in the Americas, epidemic transmission takes place during the warm season. Since its re-emergence in 1998 two major outbreaks have occurred, the biggest during 2016. To identify the environmental factors that trigger epidemic events, we analyzed the occurrence and magnitude of dengue outbreaks in time and space at different scales in association with climatic, geographic and demographic variables and number of cases in endemic neighboring countries.

**Methods**. Information on dengue cases was obtained from dengue notifications reported in the National Health Surveillance System. The resulting database was analyzed by Generalized Linear Mixed Models (GLMM) under three methodological approaches to: identify in which years the most important outbreaks occurred in association with environmental variables and propose a risk estimation for future epidemics (temporal approach); characterize which variables explain the occurrence of local outbreaks through time (spatio-temporal approach); and select the environmental drivers of the geographical distribution of dengue positive districts during 2016 (spatial approach).

**Results**. Within the temporal approach, the number of dengue cases country-wide between 2009 and 2016 was positively associated with the number of dengue cases in bordering endemic countries and negatively with the days necessary for transmission (DNT) during the previous autumn in the central region of the country. Annual epidemic intensity in the period between 1999–2016 was associated with DNT during previous autumn and winter. Regarding the spatio-temporal approach, dengue cases within a district were also associated with mild conditions in the previous autumn along with the number of dengue cases in neighboring countries. As for the spatial approach, the best model for the occurrence of two or more dengue cases per district included autumn minimum temperature and human population as fixed factors, and the province as a grouping variable. Explanatory power of all models was high, in the range 57–95%.

**Discussion**. Given the epidemic nature of dengue in Argentina, virus pressure from endemic neighboring countries along with climatic conditions are crucial to explain disease dynamics. In the three methodological approaches, temperature conditions during autumn were best associated with dengue patterns. We propose that mild autumns represent an advantage for mosquito vector populations and that, in temperate regions, this advantage manifests as a larger egg bank from which the adult population will re-emerge in spring. This may constitute a valuable anticipating tool for high transmission risk events.

# INTRODUCTION

Dengue is a mosquito-borne infectious disease caused by a virus with four serotypes (DEN 1-4) of the *Flaviviridae* family, which is transmitted to man by the bite of mosquitoes of the genus *Aedes*, mainly *Aedes aegypti* in urban areas (*Kraemer et al., 2015*). Clinical manifestations of the disease vary widely from asymptomatic to high fevers, headache, muscle and joint pain, and in some severe cases plasma leakage, hemorrhages and death (*Polwiang, 2016*). It stands as one of the main emergent tropical diseases, affecting 390 million people per year and a tenfold at risk in 128 countries, with an estimated annual global cost of US$8.9 billion (*Bhatt et al., 2013*; *Brady et al., 2012*; *Shepard et al., 2016*). The major disease burden is registered in South East Asia, South Asia and Latin America; during 2016, over 2.3 million cases were reported only in the Americas, with 244.8 cases for every 100,000 inhabitants (*PAHO, 2018*).

Where dengue is epidemic, the occurrence of an outbreak depends on virus arrival, the presence of a susceptible human population, a competent vector population and adequate environmental conditions for virus development and transmission. Therefore, dengue dynamics is affected by multiple mechanisms, in which temperature is an important determinant of mosquito traits relevant to transmission, namely the biting rate, egg and immature development, adult survival and fecundity, and development time of the virus in the mosquito (reviewed in *Mordecai et al., 2017*). While both seasonal and inter-annual climatic variability influence the geographical distribution of *A. aegypti*, other factors also determine habitat suitability. Importantly, the successful exploitation of artificial containers as larval habitats, which translates in a high domestic condition of the vector, allows *A. aegypti* to persist in regions that may otherwise be unsuitable based solely on climate (*Jansen & Beebe, 2010*). Along with local climate, El Niño Southern Oscillation has also been reported to play a role in dengue dynamics at the seasonal and inter-annual scales (e.g., *Vincenti-Gonzalez et al., 2018*).

In Argentina, *A. aegypti* is distributed along temperate and subtropical latitudes (*Vezzani & Carbajo, 2008*). Adult activity is concentrated in the warm season throughout its distribution and is absent during winter in temperate zones (*Carbajo & Vezzani, 2015*). As neighboring countries to the northeast (Brazil and Paraguay) and northwest (Bolivia)

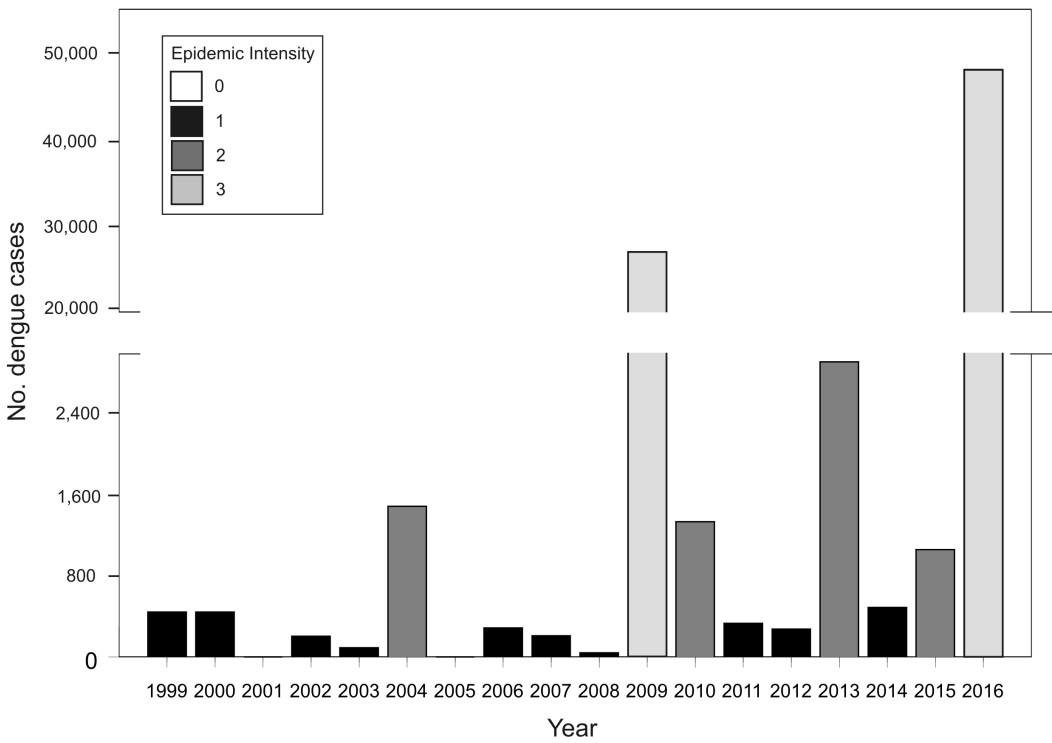

**Figure 1** Total number of dengue cases registered in Argentina per epidemiological year (defined from 1 July Year X-1 to 30 June Year X) for the period between 1999–2016, and classification of each year in epidemic intensity 0 to 3.

are endemic for dengue, Argentina represents the southern limit of dengue transmission in South America with epidemic outbreaks concentrated in the warm season (November to May) (*Carbajo, Cardo & Vezzani, 2012*). In this transmission fringe, dengue re-emerged in 1998 causing epidemic outbreaks of different magnitudes in tight association with the incidence in endemic neighboring countries (Fig. 1). Before 2016, the biggest outbreak had occurred in 2009, with nearly 27,000 cases of DEN 1, followed by 2013, in which 2,922 cases were reported and co-circulation of DEN 1, 2 and 4 was verified (*MSN, 2015*). In 2016, the biggest dengue epidemic in the country so far was experienced, concomitant with first to date autochthonous transmission of Zika and chikungunya (26 and 322 confirmed cases, respectively) (*MSN, 2016*). Also, a recent outbreak of yellow fever has been reported in Brazil with over 1,000 confirmed cases (*Ministério da Saúde do Brasil, 2018*). These arboviruses are all transmitted in urban settings by the same mosquito vector, *A. aegypti* (*Kraemer et al., 2015*). There is currently no antiviral therapy against dengue, and although the vaccine CYD-TDV "Dengvaxia®" has been approved in the country (*ANMAT, 2017*), the development of haemorragic dengue in a first infection after vaccination has been recently reported (*WHO, 2017*). Therefore, preventing contact between mosquitoes and people is still considered the main tool in the struggle against dengue.

Despite its recent history of epidemics, several studies with different geographic scopes have been performed to study dengue dynamics in Argentina. The spatio-temporal pattern

of dengue cases during the 2009 epidemic has been analyzed within a city (*Seijo et al., 2009*; *Estallo et al., 2014*) and at the country scale (*Carbajo, Cardo & Vezzani, 2012*). Regarding the 2016 outbreak, studies have been undertaken only at the city scale (*Rotela et al., 2017*; *Carbajo et al., 2018*). As no country-wide comprehensive analysis has been performed so far, and given the recent upsurge of the disease, the objective of this work was to identify the environmental factors that trigger epidemic events in Argentina by analyzing the spatio-temporal pattern of dengue cases since the re-emergence of the disease.

## MATERIALS & METHODS

Mainland Argentine extends from 22° to 55°S, encompassing subtropical and temperate latitudes (Fig. 2). Along with its neighboring countries, it is located between the two great oceans of the Southern Hemisphere, the Atlantic and Pacific Oceans. This configuration reduces daily and annual thermal amplitudes in comparison to similar latitudes of the Northern Hemisphere. The Andes Mountains, with a height range of 2,600—6,000 m.a.s.l., also greatly influences the regional climate by preventing the passage of moisture from the Pacific Ocean (*Barros et al., 2015*).

Mean temperature has increased by about 0.5 °C across most of Argentina during the past century. The strongest positive changes since 1960 occurred in the mean summer minimum temperature, even though in this season the mean maximum temperatures mostly decreased, except in Patagonia. In the second half of the past century, there was a general warming in Patagonia where both the maximum and the minimum temperature had a positive trend that was consistent with a more frequent northern flow component in the low levels of the atmosphere. Regarding precipitation, annual positive trends in northern Argentina can be partly attributed to variations in the frequency and intensity of the El Niño-Southern Oscillation (ENSO) phases. The rest of subtropical Argentina was influenced by the southern shift of the western edge of the South Atlantic high that enhanced moist advection from the Atlantic Ocean (*Barros et al., 2015*).

Information on dengue cases was obtained from dengue notifications reported in the National Health Surveillance System. During laboratory surveillance (SIVILA), diagnosis was confirmed through the detection of viral genome by polymerase chain reaction or detection of neutralizing antibodies (IgG) by plaque reduction neutralization test (PRNT). After the onset of epidemics in a given city, cases were further confirmed by detection of virus-specific immunoglobulin M (IgM) antibodies or NS1 antigen detection. Unknown data regarding department and province of residence was reconstructed using different sources: (1) district and locality in the domicile section of the SIVILA database; (2) direct consultation with the provincial Health Surveillance Area referents; or (3) data of province and district of sample collection consigned in SIVILA. Data of province or country of contagion was reconstructed from epidemiological comments of each case report. Cases registered in the SIVILA database were classified as autochthonous or imported according to the site of acquisition of the infection. All cases whose possible site of infection coincided with the jurisdiction of the patient's habitual residence, without a history of travel to an area with dengue virus circulation, were defined as autochthonous. All cases with residence in an area without dengue virus circulation and with a history of travel to an area with

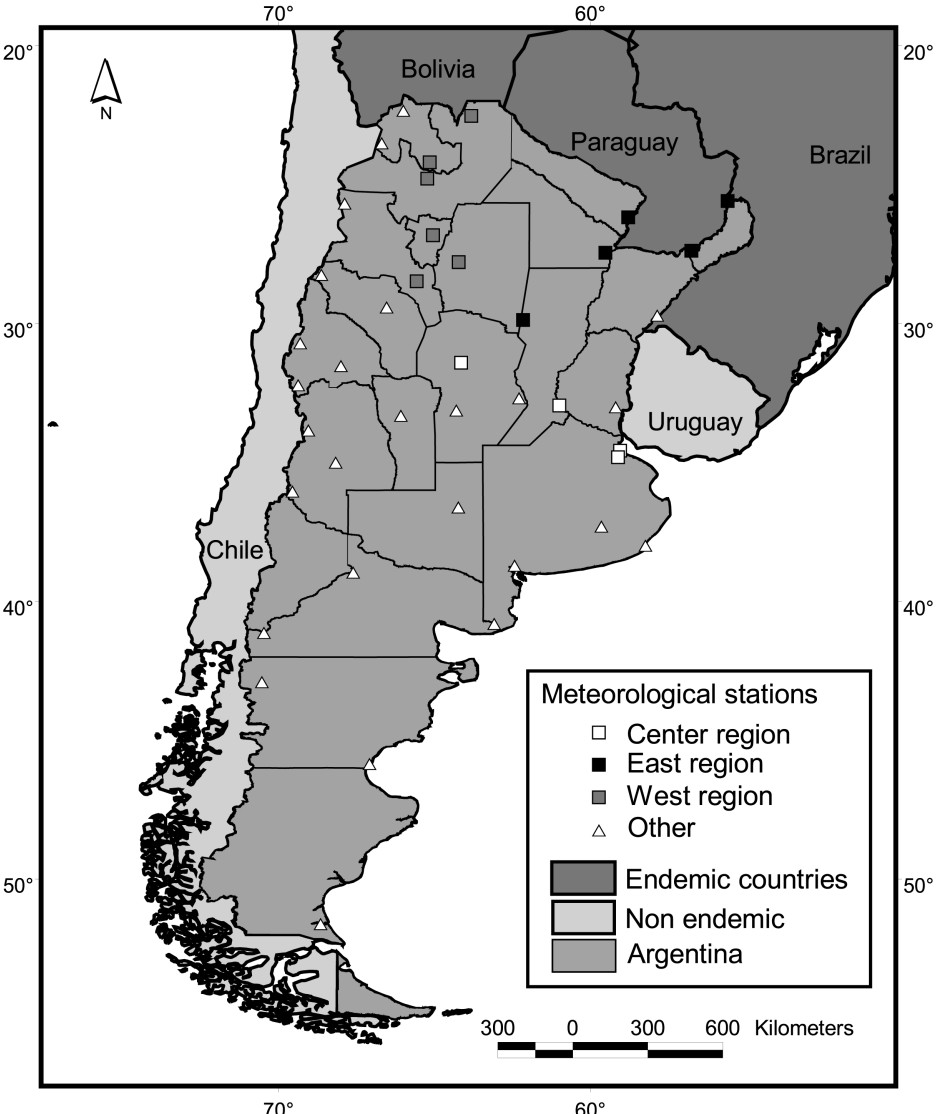

**Figure 2  Study area and meteorological stations. Squares indicate stations adjacent to districts with dengue cases in at least one year since 2009.**

circulation of the dengue virus in the last 15 days prior to the onset of symptoms were defined as imported. The database was available per district and epidemiological week.

For all analyses hereafter epidemiological years were defined as from 1 July Year 1 through 30 June Year 2 (e.g., reference to Year 2004 includes the period from 07/01/2003 to 06/30/2004). In accordance, climatic seasons were defined as follows: winter 1 Jul–30 Sep, spring 1 Oct–31 Dec, summer 1 Jan–31 Mar, autumn 1 Apr–30 Jun.

## Explanatory variables

The selection of explanatory variables was based on available data at the spatial and temporal detail required along with previous knowledge of which factors may affect dengue dynamics

**Table 1  Explanatory variables included in statistical models for temporal 2009–2016 (Ta), temporal 1999–2016 (Tb), spatio-temporal (ST) and spatial (S) methodological approaches.**

| Variable class | Variable name | Description | Units | Source | Included in approach |
|---|---|---|---|---|---|
| Climatic | Tme[*] | Mean temperature | °C | [1] | Ta - Tb - ST - S |
| | Tmi[*] | Minimum temperature | °C | [1] | Ta - Tb - ST - S |
| | PP | Mean annual cumulative precipitation | mm | [1] | Ta - Tb - ST - S |
| | DE | Mean annual dew point | °C | [1] | Ta - Tb - ST - S |
| | WI | Mean annual windspeed | m/s | [1] | Ta - Tb - ST - S |
| | DNT[*] | Days necessary for transmission | days | [1] | Ta -Tb - ST - S |
| | DPT[*] | Days of possible transmission | days | [1] | Ta -Tb - ST - S |
| | JnDe | Sum of Niño monthly index for the 12 months of the previous year (e.g., Jan-Dec 2000 for year 2001) | – | [2] | Ta - Tb - ST |
| | JnJn | Sum of Niño monthly index from January through June of the previous year. | – | [2] | Ta - ST |
| | JlDe | Sum of Niño monthly index from July–Dec of the previous year | – | [2] | Ta - ST |
| | ApSe | Sum of Niño monthly index from April–Sept of the previous year | – | [2] | Ta -Tb - ST |
| Epidemiologic | DenBol | Number of dengue cases in Bolivia | | [3] | Ta - ST |
| | DenPar | Number of dengue cases in Paraguay | | [3] | Ta - ST |
| | DenBra | Number of dengue cases in southern Brazil | | [4] | Ta - ST |
| Geographic | Ar | Area of each district | m$^2$ | [5] | S |
| | Al | Mean district elevation above sea level | m | [6] | S |
| | AlSd | Standard deviation of altitude of all pixels within a district | m | [6] | S |
| | DiWa | Distance to the nearest water body or course (excluding the sea) | Km | [5] | S |
| | DiBol | Distance to nearest border crossing to Bolivia | Km | [5] | S |
| | DiNea | Distance to nearest border crossing to Brazil/Paraguay | Km | [5] | S |
| Demographic | Pop | Population per district | people | [7] | ST - S |
| | Prc | Percentage of population change per district | – | [7] | S |

**Notes.**

*Calculated at different time spans: epidemiological year, season (winter, spring, autumn) and month. Also for each time span, regional (center, east and west) averages were calculated.

Data sources:

[1] *NCDC (2016)*
[2] *NOAA (2017)*
[3] *PAHO (2015)*
[4] *Ministério da Saúde do Brasil (2017)*
[5] *United States Geological Survey (2005)*
[6] *Instituto Geográfico Nacional (2010)*
[7] *INDEC (2017)*

(see *Carbajo, Cardo & Vezzani, 2012*). For instance, the distance to the nearest water body has been described as a proxy for the need of people to store water in containers, which eventually become larval habitats for *A. aegypti*, whereas human population increase or decrease in a given locality may reflect habitational and urbanization processes also associated with the generation of potential immature habitats. Variables included were divided in four classes: climatic, epidemiologic, geographic and demographic (definition of variables, units and data sources are shown in Table 1, Pearson's correlation coefficients between pairs of variables in File S1).

Climatic variables were calculated based on the Global Surface Summary of the Day, downloaded from NOAA Satellite and Information Service (*NCDC, 2016*). This data

is derived from the Integrated Surface Hourly (ISH) dataset, which includes global data obtained from the USAF Climatology Center and considers a minimum of four observations per day. We downloaded daily values from 33 country-wide meteorological stations throughout the country (Fig. 2), which presented no missing values for periods longer that 15 consecutive days along the period 1999–2016. To calculate country-wide annual and seasonal values, daily records of all meteorological stations located contiguous to case positive districts for at least one year were averaged (Fig. 2). We considered this subset of 15 stations because in such locations, conditions for the occurrence of cases are guaranteed, i.e., the vector is present in the district (*Vezzani & Carbajo, 2008*) and virus transmission has been reported. Also, three regions were defined following the criterion of dengue cases recompilation of the National Ministry of Health (center, east and west, Fig. 2) and values of the stations within each region were averaged. To obtain single mean values of temperature, precipitation, dew point and wind speed per district, monthly means from the period July 2011–June 2016 were averaged for each of the 33 meteorological stations. Such values were interpolated for the whole country (inverse weighted distance method, 15 km square cell grid) and the grid value corresponding to the centroid of each district was extracted and taken as the value for the district. For this interpolation, eight fictitious stations were added at the northwestern limit of the country, in the high Andes range, which were assigned the lowest values in the lowlands for each climatic variable. As no stations are present in the region, low values were needed to limit the interpolation to the west.

The extrinsic incubation period (EIP) of the dengue virus in the mosquito is the lapse from ingestion of infected blood to the virus transmission in a subsequent feed, and varies as a function of temperature (*Morin, Comrie & Ernst, 2013*). The proportion of the EIP completed per day was calculated for each meteorological station using a temperature dependent model in two hour intervals based on an asymmetric interpolation of the daily maximum and minimum (*Carbajo, Cardo & Vezzani, 2012*). The function is inhibited by low and high temperatures (around 0 and 40 °C respectively, details in File S2). With this information, two metrics were estimated. The days of possible transmission (DPT) is the number of days per year that the EIP could be completed before the death of the vector. It adds up the proportion of daily virus development for a number of days equal to the mosquito life expectancy. If unity is reached it is assigned a value of 1, and 0 otherwise (*Carbajo, Cardo & Vezzani, 2012*). This metric has the caveat of having to define a life expectancy, which was set to 15 days based on a previous study (*Carbajo et al., 2001*). A value for each district was obtained using the same methodology as for climatic variables described above. The second metric counts the number of days necessary for transmission (DNT). Beginning in each day of the year, the proportion of daily virus development is added up until unity is reached, a lower DNT value indicating higher transmission risk. It was resumed by its monthly and seasonal mean for each of the 15 meteorological stations, all values were averaged and also regional (center, east and west) values were calculated. The relation among DNT, DPT and mean temperature is shown in File S3.

The ENSO condition includes El Niño, La Niña and neutral phases. It was considered in the models by means of the monthly Oceanic Niño Index 3.4 (5N-5S, 120E-170W),

estimated as a trimester mobile mean (*NOAA, 2017*). This index, which captures SST anomalies in the central equatorial Pacific, presents a unique value for the entire country. A El Niño event occurs when this anomaly is positive (above 0.5 °C) and a La Niña event occurs when it is negative (below −0.5 °C) (*Dogliotti, Ruddick & Guerrero, 2016*). We considered the sum of the index for six and twelve months as a relative indicator of the magnitude of such events, considering different time spans and moments of the previous year (see Table 1).

Last national censuses of human population by locality were performed in 2001 and 2010 (*INDEC, 2017*). Two demographic variables were considered, namely the population number per district and the proportion of population change between both censuses, calculated as (population in 2010–population in 2001)/population in 2001 (*Carbajo, Cardo & Vezzani, 2012*).

The number of dengue cases in bordering countries with endemic transmission was obtained for the whole country in the case of Bolivia and Paraguay (*PAHO, 2018*). Given its extension, the number of cases in Brazil was calculated considering only the three southernmost districts that limit with Argentina (Santa Catarina, Rio Grande do Sul and Parana, (*Ministério da Saúde do Brasil, 2017*). As information was available at the annual scale in the traditional definition (1 January–31 December), annual cases corresponding to the current year were used (e.g., cases in 2004 for the Year 2004).

## Statistical modeling

Generalized Linear Mixed Models (GLMM) can treat data with errors that do not follow a normal distribution, and include random terms (grouping variables) to account for temporal or spatial correlation. Error distribution was selected in each analysis according to the definition of the response variable (Gaussian, Poisson or dichotomic binomial). First, univariate analyses were run (results in File S4). Then, a forward stepwise procedure was performed in which centered explanatory variables were entered one-by-one, along with quadratic relations and two-way interactions. Colinearity issues were first assessed considering correlation matrices (File S1) and further tested in every step of the modelling with the variance inflation factors (VIF, Car package). If any of the VIF values was higher than 5, which indicates multicolinearity (*Zuur, Ieno & Elphick, 2009*), the variable with the highest VIF was dropped, all VIFs were recalculated and the process was repeated until all values were lower than 5.

Once the best fixed model was achieved, the Province, District or Year (according to the scope of each methodological approach) were tested as random intercepts or slopes. Decision rules for random factor addition and variables inclusion were based on the Akaike's information criterion (AIC) (*Akaike, 1974*); the model that yielded the lowest AIC was selected from all possible models (*Zuur et al., 2009*). Graphical verification of the residuals was performed to verify the assumptions of the models.

For Gaussian and Poisson distributions, the percentage of explanation achieved was calculated as the deviance explained by the selected model divided by the deviance of the null model. For dichotomic models, as the output variable of the binomial model lies between 0 and 1, a threshold probability must be selected to distinguish positivity from
negativity (dengue occurrence and absence, respectively). All possible cut-off points from 0.01 to 0.99 were assessed to select an optimum cut-off point (cp) which maximized the classification effectiveness of the model. This was evaluated by applying the Kappa index (K) to assess improvement of classification of the model over chance (*Fielding & Bell, 1997*). Finally, for models including a random factor, variance percentages explained by the fixed (marginal $R^2$) and fixed + random (conditional $R^2$) terms were calculated (MuMIn Package).

To analyze the yearly temporality and spatial distribution of dengue cases in Argentina along the period 1999–2016, three methodological approaches were followed: (1) the *temporal approach* aimed to identify the years in which the most important outbreaks occurred in association with environmental variables and propose a risk estimation for future epidemics; (2) the *spatio-temporal approach* intended to identify which variables explain the occurrence of local outbreaks through time, taking into consideration the spatial variation country-wide; and (3) the *spatial approach* put particular emphasis in the 2016 epidemics, to describe the geographical distribution of dengue cases and identify its environmental drivers.

### Temporal approach

For this analysis, two data sets were defined:

Ta. Annual cumulative number of dengue cases at the country level between 2009 and 2016. The response variable was log ($n°$ of cases), modeled in a GLM with Gaussian error distribution (link identity) as a function of climatic variables and dengue cases in bordering countries (Table 1). This period presents the best quality in surveillance data. Before the 2009 epidemics, records had lower standards.

Tb. Annual cumulative number of dengue cases at the country level between 1999 and 2016. Acknowledging different surveillance data quality, the analysis of this dataset attempted to predict the risk of future outbreaks including all available information. Given that the number of cases presents a random component and too much dispersion, the response variable was classified in four categories of epidemic intensity: 0 (0 cases), 1 (1–1,000), 2 (1,001–5,000), 3 (>5,000) (Fig. 1). The model presented a Poisson error distribution and log link. Explanatory variables describing El Niño events during the summer and dengue cases in bordering countries were excluded because the time span in which they are calculated restrict anticipating power.

### Spatio-temporal approach

From a database available per week at the district level, we considered cases between 2009 and 2016 grouped by district and epidemiological year. This analysis was restricted to case positive districts for at least one year and contiguous to any of the available 33 meteorological stations. The combination of both conditions resulted in a subset of 15 districts retained (squares in Fig. 2). Two alternative dichotomic response variables were defined. Of the first, a value of 1 was assigned to a district in a given year if >20 cases were reported that year, a value of 0 otherwise (30% of the database is classified as positive; the number of positive district does not differ when considering the threshold between 15 and 20 cases). In the second, a value of 1 was assigned to a district in a given year if >100 cases
were reported, a value of 0 otherwise (20% of the database classified as positive). Both were modeled in a GLM with binomial error distribution and link logit. Explanatory variables included were climatic and cases in endemic neighboring countries (Table 1).

### Spatial approach

The database consisted in the number of cases (between epidemiological weeks 43 of 2015 and 29 of 2016) per district for the whole country (503 districts in total). The response variable was the occurrence of two or more dengue cases per district (dichotomic) in a binomial GLM with logit link. Four districts with up to two reported cases located southwards of the geographical distribution of *A. aegypti* were deliberately excluded assuming a potential mistake in the classification as autochthonous. Explanatory variables included were climatic, demographic and geographic (Table 1).

## RESULTS

### Temporal approach

The best model associated the (log transformed) number of dengue cases at the national scale between 2009 and 2016 (Ta) positively with the number of dengue cases in bordering endemic countries and negatively with the DNT during the autumn of the previous year in the central region of the country (Table 2). In other words, higher number of cases in Bolivia and south of Brazil were correlated with more intense outbreaks in Argentina, and when a given autumn presented fewer days necessary for transmission, a larger outbreak could be expected the following warm season. The latter result makes sense for the central temperate region, in which temperature is a limiting factor for vector development, whereas in northern subtropical areas DNT values are consistently smaller. The model explained 95.2% of the variability of the data, and the correlation between the response values and those predicted by the model was 0.97. However, the model required three parameters for explaining only eight values. No retro extrapolation of this model to the years 1999–2008 could be performed, given that no reports of dengue cases in Bolivia were available for those years.

When aiming to propose a model able to predict future dengue epidemics, considering the entire time span of epidemics since vector re-invasion (Tb) and deliberately excluding variables with no predicting power, the best model associated the intensity of epidemics with temperature conditions during the previous autumn and winter, estimated as days of possible transmission for each season (Table 2). Although at lower explanation percentages (57%), once again the meteorological situation during the previous autumn was a significant predictor of the following outbreak. This model is valuable in providing a risk estimation for the following warm season based on in advance and easily available climatic data as only input. The model predicted an intensity epidemic of class 1 for 2017, as has been verified to date (254 confirmed and 299 probable cases up to epidemiological week 28; *MSN, 2017*).

### Spatio-temporal approach

When considering >20 cases as an epidemic, the selected model classified the observed values 72% better than chance ($K = 0.72$), at a cut-off point of 0.38. None of the tested

**Table 2  Selected models for the different methodological approaches used to study the environmental and demographical determinants of dengue epidemics in Argentina.**

| Methodological approach | Response variable | Explanatory variable | Estimate ± standard error |
|---|---|---|---|
| Temporal 2009–2016 | Log (n° cases) | Ordinate | $7.59 \pm 0.20$*** |
| | | $DNT_{autumn\text{-}center}$ | $-1.11 \pm 0.25$* |
| | | DenBol | $1.85 \pm 0.29$** |
| | | DenBra | $1.29 \pm 0.32$* |
| 1999–2016 | Epidemic intensity | Ordinate | $-0.01 \pm 0.28$ |
| | | $DNT_{autumn}$ | $-0.78 \pm 0.28$** |
| | | $DNT_{winter}$ | $0.47 \pm 0.23$* |
| Spatio-temporal | Occurrence > 20 cases | Ordinate | $-1.41 \pm 0.31$*** |
| | | $DNT_{may}$ | $-0.29 \pm 0.09$** |
| | | DenBol | $0.07 \pm 0.02$*** |
| | | $DNT_{spring}$ | $0.33 \pm 0.13$** |
| | | DenPar | $0.024 \pm 0.007$*** |
| | | $DNT_{may}$: DenBol | $0.008 \pm 0.003$** |
| | Occurrence > 100 cases | Ordinate | $-2.35 \pm 0.45$*** |
| | | $DNT_{may}$ | $-0.29 \pm 0.10$** |
| | | DenBol | $0.09 \pm 0.02$*** |
| | | DenBra | $0.06 \pm 0.02$*** |
| | | Pop (in thousands) | $0.0008 \pm 0.0004$* |
| | | $DNT_{may}$:DenBol | $0.006 \pm 0.002$* |
| Spatial | Occurrence ≥ 2 cases | Ordinate | $-2.33 \pm 0.51$*** |
| | | $Tmi_{autumn}$ | $2.88 \pm 0.46$*** |
| | | Log(Pop) | $2.49 \pm 0.32$*** |
| | | (1|Province) | |

**Notes.**

Asterisks next to the values indicate statistical significance *$p < 0.05$, **$p < 0.01$, ***$p < 0.001$.
See variables abbreviations in Table 1.

grouping random factors (1|District, 1|Year and Year|District) significantly improved the model. DNT values in autumn months of the previous year (best represented by May) resulted determinant; once again, dengue cases within a district were associated with mild conditions during the previous autumn (that is, lower DNT values). High DNT during the spring also favored the occurrence of epidemics, along with the number of dengue cases in Bolivia and Paraguay (Table 2).

When considering >100 cases as an epidemic, the selected model classified the observed values 63% better than chance ($K = 0.63$), at a cut-off point of 0.39. No random factors were significant. Results were consistent with the former analysis and associated the occurrence of an epidemic with the DNT of the previous May, the intensity of outbreaks in neighboring countries (Bolivia and Brazil), and positively with human population density (Table 2).
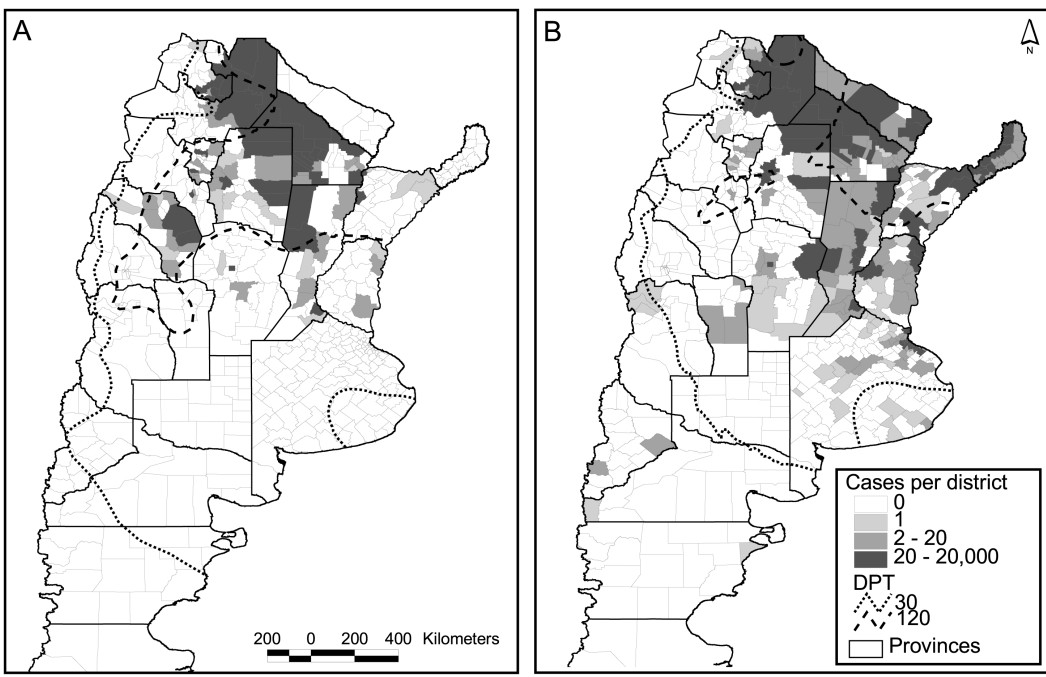

**Figure 3** Dengue cases per district and days of possible transmission (DPT) isolines for 2009 (A) and 2016 (B).

## Spatial approach

Out of the 503 districts in the country, 272 were negative for dengue cases in both major epidemics (2009 and 2016), whereas 73 reported one or more cases in both occasions. A strong increase in the number of positive districts was verified in 2016 compared to 2009, with 133 new districts reporting cases, and only 25 with the opposite behavior (Fig. 3). The relation autochthonous/imported cases was, however, very similar in both years, around 26 autochthonous cases per imported case (47,894/1,850 for 2016 and 26,923/~1,000 for 2009). Although not reflected in the number of cases and districts affected, considering temperature conditions expressed in terms of days of possible transmission (DPT) the situation in 2009 was worse than in 2016 (isolines of 30 and 120 DPT spread further south and west, Fig. 3).

The best model for the occurrence of two or more dengue cases per district included autumn minimum temperature and human population (log scale) as fixed factors, and the province as a random grouping variable (Table 2). Mild temperatures during the previous autumn and higher population numbers were associated with the occurrence of dengue cases per district. Prediction accuracy of the observed values was 80% better than chance ($K = 0.8$) at a cut off-point of 0.4. Explanatory power of the fixed factors (marginal $R^2$) was 0.74, whereas including the random factor (conditional $R^2$) raised explanation to 0.86.

The model published in *Carbajo, Cardo & Vezzani (2012)* for the 2009 epidemics was re-run using the occurrence of two or more dengue cases per district in 2016 as a response variable, and the same explanatory variables that predicted the previous outbreak. These

were the DPT, distance to water courses, population number (log scale) and the percentage of population change in the period 2001–2010 (last two national censuses). In Argentina, national censuses are typically performed on a decadal basis; therefore, updating of the demographic variables was unfeasible. The model was not very accurate in predicting the new cases ($K = 0.47$, correct classification 0.7, sensibility 0.4, specificity 0.9). An alternative explanation for the low predictive model is the difference in the origin of both epidemics. The one in 2009 came down from Bolivia, whereas the latter 2016 outbreak was caused by an unusually high incidence in north-eastern neighboring countries (Brazil and Paraguay). Also, unexpectedly two northeastern provinces did not communicate any cases during 2009 (Fig. 3A).

Conversely, the new model was re-run with the 2009 epidemics data. In this case, the predictive accuracy was high ($K = 0.71$ at a cut off point of 0.35), however the fraction explained by the fixed factors was significantly reduced (marginal $R^2 = 0.45$), whereas the variance explained by the Province as a random grouping increased (conditional $R^2 = 0.84$).

## DISCUSSION

Modeling disease outbreaks in time and space is a powerful tool to qualitatively predict future potential epidemics (*Myers et al., 2000*). Given the epidemic nature of dengue in Argentina, virus pressure from endemic neighboring countries along with climatic conditions are crucial to explain disease dynamics. Remarkably, in all three methodological approaches used to analyze the spatial and temporal pattern of dengue cases, different proxies for temperature of the previous autumn were among the best predictors. The number of days necessary for transmission presented higher explanatory power than the raw variable in two approaches. Although this proxy is mainly used for transmission risk calculations, it may well resume conditions for the mosquito. As temperatures seldom reach extreme inhibition values during autumn, the temporal detail and the inclusion of temperature amplitude in its calculation may favor DNT instead of mean temperature.

We built a temporal model based on the period 2009–2016, which lacks predictive power because it relies on the ongoing year virus pressure from endemic neighboring countries. However, it does provide a hint regarding the conditions of the previous autumn. If the autumn of a given year is propitious for transmission, caution should be taken for the following summer and the situation in bordering endemic areas should be followed up closely. We propose that mild autumns represent an advantage for mosquito vector populations. In temperate regions with no mosquito reproduction during winter, this advantage could manifest as a larger egg bank deposited during autumn from which the adult population will re-emerge in spring.

Including information of the years 1999–2008, we reached a model only relying on in advance climatic conditions. In this way, with easy-available temperature data we can anticipate if the following year will present an epidemic of mid-high intensity. Models predicting epidemic intensity rather than number of cases are more robust to particular and/or stochastic conditions of a given year. The presented model properly predicted the

2017 situation; however as only two major outbreaks (2009 and 2016) have occurred so far in Argentina to train the models, more time is needed to test its efficiency.

Spatio-temporal models of dengue cases distribution between 2009 and 2016 presented significant association with the same type of variables, prioritizing the days necessary for transmission during autumn months and the incidence in neighboring endemic countries. These models also lack predictive power and are particularly sensitive to the characteristics of the on course epidemics. This could also be appreciated when comparing 2009 and 2016 spatial models, in light of the poor correspondence in terms of numbers and geographical distribution of positive districts that presented the *Carbajo, Cardo & Vezzani (2012)* model for the 2009 epidemics to predict the situation in 2016. Both main dengue outbreaks in Argentina started in locations close to endemic borders and further expanded to other localities, however their origin was different. In 2009, the epidemic was triggered by high incidence in Bolivia, spreading south from the northwest, whereas in 2016 it started in the northeast, due to high incidence in south Brazil and Paraguay. For this reason, the proper identification of imported cases is especially relevant during transmission onset. We also found no relation between El Niño events and dengue patterns. It has been described that river flow of water courses located within the northeast region of the country are strongly affected by ENSO events, whereas those located in the northwest region are not (*Pasquini & Depetris, 2007*). This may partly explain the lack of association of dengue outbreaks and El Niño in Argentina, as described elsewhere, however we were unable to test this hypothesis due to the short history of northeast and northwest derived epidemics in the country.

The geographic distribution of dengue-positive districts during the 2016 epidemics was positively associated with minimum autumn temperature and human population, the latter variable being also associated with the 2009 outbreak (*Carbajo, Cardo & Vezzani, 2012*). A larger human population number in a district may reflect higher virus pressure due to traveling. Also, larger cities usually present precarious and overcrowded conditions, and are therefore more susceptible to experiencing an outbreak (*Brunkard et al., 2007*). This is combined with a large number of artificial containers due to uncontrolled urbanization and land uses linked to urban areas (e.g., cemeteries, rubbish and tire deposits), which represent optimum habitats for immature vector development. Within the spatial model, the inclusion of Province as a random grouping factor may reflect non biological features, such as asymmetries in case records or in the efficiency of prevention and control interventions. The total absence of cases reports during 2009 in some northeastern provinces is an extreme example of this issue. These areas experienced unreported transmission, which could have substantially affected the predictability of the 2009 model as it occurred to *Aström et al. (2012)*.

Comparing dengue distribution models may be problematic because of differing modeling approaches and scales, the diverse quality of the data used, and the selection of variables associated with disease distribution (*Messina et al., 2015*). Further issues arise in the particular case of Argentina due to its geography. It constitutes one of the potential areas of transmission expansion and intensification, where many models do not agree in their predictions (*Messina et al., 2015*). It extends along 3,800 km including a gradient of transmission risk due to climatic variables roughly decreasing from north to south, but

almost a third of the population lives at the transmission fringe in the center-east of the country. The epidemic inter-annual pattern is irregular, with frequent virus circulation in the north and large sporadic outbreaks extending to almost half of the country. This implies that the maximum geographic extension does not repeat regularly and may respond to particular conditions occurring in certain years. Global models are not aimed at identifying these particular events, what would need a spatial and temporal detail not yet implemented or simply not available globally. Although the inclusion of demographic and socio-economic factors has improved global models (e.g., *Aström et al., 2012*; *Mordecai et al., 2017*), its application in Argentina would require a district detail to account for the internal heterogeneity in, for example, the geographical distribution of income along the country.

The systematic data gathering of dengue cases in a country-wide system allows for the constant updating of the epidemiological situation, as well as the identification of high risk areas to prioritize prevention and control measures both at the communitarian and individual levels. The strength of the 2016 outbreak manifests the flaws of the mitigation strategies. This, in conjunction with a higher dengue incidence in neighboring countries and the introduction of other arboviruses transmitted by the same vector (Zika, chikungunya, and potentially yellow fever), forecast a complex scenario which claims for an integrative research approach with joint work from the areas of education, infrastructure and health. Along with the need to develop better and faster diagnostic methods, we need to achieve consciousness in the population, whose contribution is essential. The optimal use of models to inform policy decisions requires a continuous dialogue between the multidisciplinary infectious disease dynamics community and decision makers (*Heesterbeek et al., 2015*), essential to plan and put into practice adequate prevention and control measures for each jurisdiction.

## CONCLUSION

In the southernmost extreme of epidemic transmission in the Americas, the pattern of dengue cases concentrated during the warm season was associated with different proxies for temperature conditions during the previous autumn and virus pressure from neighboring endemic countries. This provides a valuable opportunity to prepare in advance for a qualitatively strong epidemic during the following summer if the autumn of a given year is propitious for transmission, and enable the onset of early alerts by close follow up of the situation in endemic bordering countries.

### Funding
This work was supported by Becas SALUD INVESTIGA ''Dr. Abraam Sonis'', categoría ESTUDIO MULTICENTRICO, Dirección de Investigación para la Salud, Ministerio de Salud de la Nación. The funders had no role in study design, data collection and analysis, decision to publish, or preparation of the manuscript.

## Grant Disclosures

The following grant information was disclosed by the authors:

Becas SALUD INVESTIGA ''Dr. Abraam Sonis'', categoría ESTUDIO MULTICENTRICO, Dirección de Investigación para la Salud, Ministerio de Salud de la Nación.

## Competing Interests

The authors declare there are no competing interests.

## Author Contributions

- Anibal E. Carbajo conceived and designed the experiments, analyzed the data, prepared figures and/or tables, authored or reviewed drafts of the paper, approved the final draft.
- Maria V. Cardo analyzed the data, prepared figures and/or tables, authored or reviewed drafts of the paper, approved the final draft.
- Pilar C. Guimarey and Arturo A. Lizuain performed the experiments, approved the final draft, database gathering and revision.
- Maria P. Buyayisqui and Teresa Varela performed the experiments, contributed reagents/materials/analysis tools, approved the final draft.
- Maria E. Utgés performed the experiments, authored or reviewed drafts of the paper, approved the final draft, database gathering and revision.
- Carlos M. Giovacchini conceived and designed the experiments, performed the experiments, contributed reagents/materials/analysis tools, approved the final draft.
- Maria S. Santini conceived and designed the experiments, performed the experiments, approved the final draft, group coordination.

## Data Availability

Carbajo, Anibal E.; Cardo, Maria V; Guimarey, Pilar C; Lizuain, Arturo A; Buyayisqui, Maria P; Varela, Teresa; Utgés, Maria E; Giovacchini, Carlos M; Santini, Maria S (2018): Dengue cases in Argentina at different spatial at temporal spans. figshare. Dataset. https://doi.org/10.6084/m9.figshare.6635606.v1.

## Supplemental Information

Supplemental information for this article can be found online at http://dx.doi.org/10.7717/peerj.5196#supplemental-information.

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
