# Peer review of "Is autumn the key for dengue epidemics in non endemic regions? The case of Argentina"

_PeerJ, doi:10.7717/peerj.5196_

## Round 0.1 · original submission · Major Revisions

Please pay particular attention to technical comments from the reviewer 2. Many clarifications about the statistical model are required.

Reviewer 1 ·

Basic reporting

Generally, this manuscript was presented in well written English. But in many places, compound sentences with attributive clauses are ambiguous, better to be checked for syntactical corrections. The following are some examples:
Line 30-32: syntactical error
Line 49-51: please reconstruct this sentence.
Line 80-82: confusing sentence
Line 98-100: syntactical error
Line 174-176: syntactical error
Line 231: in → of

Literature references were sufficiently presented and article structure was organized clearly. About supplementary materials, it would be more informative for readers if the case data are included in the supplementary part. And in currently existing supplementary txt, necessary explanation for abbreviations should be given as some are not shown in the Table 1.

Experimental design

This study is trying to explore the environmental factors that drive epidemic events in Argentina by employing well collected meteorological data from country-wide stations and annual dengue cases registered in SNVS. Good models can help forecast future epidemic outbreak. This topic is important in the field of epidemic, and author’s attempt is appreciable. This article involved no ethical issue, and conformed to PeerJ scope completely.

Technically, this article is a continuation of one published paper as referenced [Carbajo, Cardo & Vezzani, 2012]. Strict statistical analysis was conducted aiming to obtain optimum models which are capable of describing most contributive factors for dengue cases. Some details in the Materials & Methods need to be clarified.

Line 134-137: Using climatic data from 15 meteorological stations, interpolation for the whole country was conducted based on 15 km square cell grid. The sensitivity analysis of models to meteorological station data is necessary, considering the accuracy and reliability of sparse data including eight artificial stations.

Line 142: While definition of DPT was oriented in reference (Carbajo, Cardo & Vezzani, 2012), explicit presentation of DNT, i.e., mathematical relationship between DNT and temperature variable is essential in this article. Because following results regarding DNT and DPT were showed alternatively, comparative differences between the two terms need to be illustrated.

Validity of the findings

Selected models according to three approaches are of high power to explain response variable, i.e., dengue cases. Following points are expected to complete model analysis.

1) Considering final results were presented with selected mathematical models according to three statistical approaches, more explicit comparisons among candidate models should be given. For instance, in Line 342-343, authors summarize no association with El Niño, but in prior part, there are no analysis involving El Niño events as explanatory variable.
2) Line 305: 2009 epidemic model was rerun for 2016 epidemic, but showing low predictive power. The only explanation that different source countries of infection is not enough, which cannot manifest the epidemic nature.

Additional comments

Authors did a good job to identify environment factor closely related with dengue cases. Statistical approached were well designed and brief results were presented. However, in order to make this article more readable, written English is better to be improved.

Reviewer 2 ·

Basic reporting

Overall well written with few minor wording errors

Raw data have been shared.

Introduction:
L56 Briefly describe the clinical presentation of dengue virus infections.
L59 Note that the large increase in dengue cases in 2015 was due in large part to the CHIKV epidemic. Many CHIKV cases were misdiagnosed as DENV. I suggest picking a different time period for comparison.
L67 How many cases of ZIKV in Argentina? Imported or locally acquired?
L69 Correction: "an outbreak of YF was reported in the Amazon basin in 2017 with [number] of cases."
L79 Describe how/why climate affects arbovirus transmission and include relevant references from lab and field studies. Provide relevant background on the behavior of the mosquito vector.
L80 Describe what is meant by the "close association"
L82. Climate variability at what time scale? Seasonal, interannual, decadal, etc.
L85 El Niño and La Niña events have also been associated with dengue and malaria transmission in Latin America (please cite relevant literature).
L90. Update the reference: Carbajo et al in press.

Figures/Tables:
Table 2: Instead of a table, you could show a bar graph with a series of the annual cases and use different symbology in the bars to show the different categories of transmission intensity.
Table 3: For each model, please show parameter estimates, SE, p values
Figure 1. Please clarify in the figure legend how the sites were selected where the squares are located. What does "other" refer to? Please provide more detail in the figure legend.

The discussion lacks a robust interpretation of the results in comparison to the findings from other relevant studies, other than those by the authors. Please add references.

Experimental design

Major comments:
L 123 Did you analyze how the independent variables are correlated? It would be helpful to see a correlation matrix as supplementary information.
L132 Explain why these dependent variables were selected (e.g., distance to the nearest water body and % of population change). Did you examine maximum temperature and relative humidity?
L134 What is meant by the following “average records from the period July 2011- June 2016 were used to smooth distinctive and/or anomalous features of the year before and during the epidemic.” What smoothing function was used? Normally you calculate anomalies using the entire climatological time series (ideally +10 years), rather than 1 year before or after.
L144 Although the model was described previously (Carbajo et al 2012) please provide basic information about the mathematical model, including formulas and parameters. This is an important component of this paper and should be well understood. How has the model been updated/advanced since the last publication? Does the model consider the nonlinear response of the mosquito vector and viral replication in relation to temperature? See recent work by Mordecai et al 2017 PLOS NTD and other authors. Consider showing how the DNT and DPT vary with temperature.
L154 Please define El Niño events and the Niño 4. Why was this value used instead of the Niño3.4, which is more commonly used and forecasted?
L157 Why would you sum the Niño monthly index? I don't understand these variables. Also, you use the annual time period (Jan-Dec), whereas in the analysis you define the annual time period as July-June. Please be consistent.
L165 By using dengue cases from the neighboring countries from the normal calendar year (Jan-Dec), this automatically introduces a temporal lag into the analysis, since the normal calendar year begins 6 months before the beginning of the year in the analysis (July).
L202 Did you test any temporal lags of the dependent variables with respect to dengue counts?
L211, 216 (and elsewhere in the paper). The models developed here are not forecast models. If you are forecasting (predicting dengue transmission in future time periods), please describe the methods. Are you leaving out years in the time series and predicting those years using the rest of the time series? Are you using seasonal climate forecasts? etc. If yes, show time series with the forecast results.
L 213, L224, L267 Number of disease cases should be divided by population to capture disease incidence (e.g., 100 cases per 10,000 people). By not correcting for population, human population becomes a predictor in the model, which is self evident (we have more cases where we have more people in areas suitable for transmission) (L292)
L213 How were these 4 categories for transmission intensity selected? I suggest using quartiles (25/50/75th %) for the categories.
L267 Why was this threshold selected?

Minor comments:
L104. Please briefly describe the climatology of the study area. What are the key macro-climatic drivers of interannual variability in rainfall and temperature? (e.g., ENSO events or other phenomena)
L105. What diagnostic tests are used to confirm the cases by laboratory surveillance?
L 125 Please provide more information about the specific climate product used in this analysis. I could not find this information on the website provided.
L130 What data were used to confirm that the vector was present in the district? Please include this information.
L130 How were the 3 regions delimited/identified?
L237 In the methods, please describe the source of the demographic data and the datasets that were used.
L256. The authors indicate that they deliberately excluded variables with no forecasting power in the model. Which variables were included and why?
L304. In order to estimate annual population, you could use a linear extrapolation between census years, allowing you to project population based on prior trends in population growth.

Validity of the findings

The model and data sources require more description in order to understand and assess the validity of the results of the analysis.

Additional comments

The authors present a spatial and temporal analysis of dengue transmission in Argentina in relation to climate conditions and transmission in neighboring countries. The most important contribute of the paper is the identification of the autumn season as a key period regulating dengue transmission and the importance of dengue transmission in neighboring endemic countries. Although the paper addresses an important topic, it is not publishable in its current form. Specific recommendations are identified above.

---

## Round 0.2 · accepted · Accept

Congratulations. Both reviewers indicate that you have successfully addressed earlier comments.

Reviewer 1 ·

Basic reporting

no comment

Experimental design

no comment

Validity of the findings

no comment

Additional comments

Authors have addressed all my comments. Necessary information had been added; there are no further comments on the manuscript.

Reviewer 2 ·

Basic reporting

No comment

Experimental design

No comment

Validity of the findings

No comment

Additional comments

The authors have done a great job with the revisions. I have no further comments.